# Alkaline pH, Low Iron Availability, Poor Nitrogen Sources and CWI MAPK Signaling Are Associated with Increased Fusaric Acid Production in *Fusarium oxysporum*

**DOI:** 10.3390/toxins15010050

**Published:** 2023-01-06

**Authors:** Davide Palmieri, David Segorbe, Manuel S. López-Berges, Filippo De Curtis, Giuseppe Lima, Antonio Di Pietro, David Turrà

**Affiliations:** 1Department of Agricultural, Environmental and Food Sciences, University of Molise, 86100 Campobasso, Italy; 2Departamento de Genética, Campus de Excelencia Internacional Agroalimentario ceiA3, Universidad de Córdoba, 14014 Córdoba, Spain; 3Department of Agricultural Sciences, Università di Napoli Federico II, 80055 Portici, Italy; 4Center for Studies on Bioinspired Agro-Enviromental Technology, Università di Napoli Federico II, 80055 Portici, Italy

**Keywords:** *Fusarium oxysporum*, fusaric acid, pH, iron limitation, chelating activity, signaling

## Abstract

Fusaric acid (FA) is one of the first secondary metabolites isolated from phytopathogenic fungi belonging to the genus *Fusarium*. This molecule exerts a toxic effect on plants, rhizobacteria, fungi and animals, and it plays a crucial role in both plant and animal pathogenesis. In plants, metal chelation by FA is considered one of the possible mechanisms of action. Here, we evaluated the effect of different nitrogen sources, iron content, extracellular pH and cellular signalling pathways on the production of FA siderophores by the pathogen *Fusarium oxysporum* (*Fol*). Our results show that the nitrogen source affects iron chelating activity and FA production. Moreover, alkaline pH and iron limitation boost FA production, while acidic pH and iron sufficiency repress it independent of the nitrogen source. FA production is also positively regulated by the cell wall integrity (CWI) mitogen-activated protein kinase (MAPK) pathway and inhibited by the iron homeostasis transcriptional regulator HapX. Collectively, this study demonstrates that factors promoting virulence (i.e., alkaline pH, low iron availability, poor nitrogen sources and CWI MAPK signalling) are also associated with increased FA production in *Fol*. The obtained new insights on FA biosynthesis regulation can be used to prevent both *Fol* infection potential and toxin contamination.

## 1. Introduction

*Fusarium oxysporum* comprises a cosmopolitan complex of fungal species [1] including both non-pathogenic and pathogenic forms [2], which can infect plants, animals and humans [3]. Plant pathogenic strains cause tracheomycosis or foot and root rots (Fusarium wilt) in a large number of plant species and are grouped into over 150 pathogenic forms (formae speciales) [4]. *F. oxysporum* f. sp. *lycopersici* (*Fol*) is the pathogenic form that causes wilting of tomato plants. *Fusarium* species are known to synthesize several biologically active compounds with different roles in plant pathogenesis and microbial competition [5]. Fusaric acid (5-butylpyridine-2-carboxylic acid) (FA), the first fungal metabolite discovered to be implicated in tomato pathogenesis [6], is one of the most widely distributed mycotoxins in the genus *Fusarium* and has been used as an efficient indicator of *Fusarium* contamination in food and feed grains [7]. Plant treatment with FA causes the rapid development of disease symptoms such as internerval necrosis and foliar desiccation even in the absence of the pathogen [8]. FA toxicity in plants has been attributed to different mechanisms of action, including direct membrane damage, electrolyte loss, decrease in cellular ATP levels, metallo-enzyme inhibition, oxidative burst and metal chelation [6,9,10,11]. In addition to its phytotoxic effect, FA also shows varying degrees of inhibitory activity on rhizobacterial populations. For instance, species of the genera *Bacillus* and *Paenibacillus* are susceptible, while those belonging to *Pseudomonas* are highly resistant [12]. Intriguingly, FA resistance in these species has been shown to rely on the expression of two major siderophores, pyoverdine and enantio-pyochelin, suggesting that the inhibitory effect of FA toxicity on bacteria, similar to plants, depends on its chelating activity [13]. 

In *Fusarium fujikuroi* the expression of FA biosynthesis genes has been shown to be regulated by the nitrogen-responsive GATA transcription factor AreB and the pH-responsive transcription factor PacC [14]. Similarly, FA levels in *Fol* are controlled via PacC-mediated modulation of chromatin condensation at the *fub1* locus, which encodes a major FA biosynthetic gene [15]. Additionally, in a banana pathogenic isolate of *F. oxysporum* f. sp. *cubense,* several components of the CWI MAPK cascade have been shown to act as positive regulators of FA biosynthetic genes and FA production [16]. 

In *Fol*, three different MAPKs (Fmk1, Mpk1 and Hog1) regulating distinct virulence functions have been described. While Fmk1 is essential for invasive growth and plant infection, the other two MAPKs, Mpk1 and Hog1, contribute to plant infection both via Fmk1-shared and -independent functions, albeit to a lesser extent [17,18,19]. This work aimed at evaluating the role of different environmental factors such as nitrogen source, extracellular pH and iron content in the regulation of FA production in *Fol*. Furthermore, we tested the contribution of the three MAPK pathways as well as the iron and pH response regulators HapX and PacC, respectively, on FA biosynthesis. 

## 2. Results

### 2.1. Nitrogen Source Affects Extracellular pH in F. oxysporum f. sp. lycopersici

We tested the effect of different nitrogen sources on extracellular pH modification, iron chelating activity and FA production by *Fol.* Microconidia were inoculated in minimal medium (pH 4.5) supplemented with either urea, sodium nitrate, ammonium sulphate or ammonium nitrate as the sole nitrogen source. While urea and sodium nitrate elicited an increase in extracellular pH, ammonium sulphate and ammonium nitrate induced extracellular acidification (Table 1; Figure 1). To investigate the role of MAPK pathways in pH modulation, we measured extracellular pH changes in *fmk1*Δ, *hog1*Δ and *mpk1*Δ knock-out mutants. Interestingly, deletion of the cell wall integrity (CWI) MAPK Mpk1 resulted in increased extracellular alkalinisation on urea and sodium nitrate but did not affect ammonium-dependent acidification. Similarly, mutations in the seven-transmembrane α-pheromone receptor Ste2 or the conserved components of the CWI pathway Bck1 and Mkk2 also led to an increase in pH (Table 1), suggesting for a role of the sex pheromone perception machinery and the CWI pathway in the regulation of this process. 

### 2.2. Siderophore Production Is pH- and Iron-Sensing Dependent

To detect siderophore production by *Fol*, a CAS assay was performed after 5 days of incubation in the different test conditions. We found that the chelating activity in fungal supernatants was more than 40% higher in media containing urea or sodium nitrate showing high pH, compared to those supplemented with ammonium sulphate or ammonium nitrate which had low pH (Table 1). Furthermore, chelating activity was significantly higher (*p* ≤ 0.05) in isogenic mutants lacking Ste2 or conserved components of the CWI MAPK pathway, but not in *fmk1*Δ and *hog1*Δ mutants, compared with the wild-type strain (Table 1). The correlation between pH signalling and siderophore production was further supported by the finding that deletion the of alkaline pH-responsive factor PacC (*pacC*Δ) resulted in decreased chelating activity in urea and nitrate media, whereas expression of a dominant activating PacC allele (*pacC^c^*), which represses acidic-regulated functions [20,21], resulted in increased chelating activity in ammonium-containing media (Table 1).

Previous studies revealed that deletion of the iron homeostasis regulator HapX induces a slight increase in extracellular chelating activity under iron-limiting conditions when using glutamine (Gln) as the nitrogen source [22]. Here, we detected a significant increase in extracellular chelating activity in *hapX*Δ as compared to the wild type, which was even more dramatic when ammonium was used as a nitrogen source. 

### 2.3. Fusaric Acid Production Is Regulated by Environmental pH and Iron Availability 

Fusaric acid (FA) has been suggested to function in metal cation chelation [13,15]. Here, we found that FA concentrations increased steadily in supernatants of the wild-type strain grown on urea or sodium nitrate, whereas no FA was detected in cultures supplemented with ammonium even after prolonged incubation (Figure 1 and Figure 2; Table 1). These findings suggest that either the nitrogen source or environmental pH regulates FA production. In line with the second hypothesis, buffering the pH to 4.5 completely abolished FA production on urea- or sodium nitrate-containing media, while increasing the pH to 7.0 activated FA production on ammonium-supplemented media (Figure 3). Further corroborating the finding of a pH-dependent FA regulation mechanism in *Fol*, inappropriate pH sensing in the *pacC*Δ and *pacC^c^* mutants resulted in altered FA levels at alkaline or acidic pH, respectively. 

In general, FA production appeared to correlate with chelating activity in fungal supernatants, particularly in sodium nitrate-supplemented cultures, which contained high levels of chelating activity and FA (Table 1). Interestingly, the addition of the general ion chelator EDTA or the specific iron-chelating bacterial siderophore pyoverdine resulted in increased FA concentrations regardless of the nitrogen source used, even though pyoverdine stimulation was most effective on sodium nitrate (Figure 3). By contrast, iron addition dramatically decreased FA production in all tested conditions (Figure 3), suggesting that iron sensing plays an important role in the regulation of FA biosynthesis. Indeed, the deletion of the iron-response transcriptional regulator HapX led to increased FA production in all tested media except for that containing sodium nitrate (Table 1).

Unexpectedly, the *ste2*Δ, *bck1*Δ, *mkk2*Δ and *mpk1*Δ mutants showed lower levels of FA production in both urea- and sodium nitrate-supplemented media, despite the higher chelating activity detected in these conditions (Table 1). 

## 3. Discussion

Since its discovery more than 70 years ago, fusaric acid (FA) has been among the most studied fungal secondary metabolites produced by *Fusarium* phytopathogenic species [7,23]. Its wide spectrum toxicity towards plants, animals, bacteria and fungi has attracted the attention of many scientists with the aim of identifying its biosynthetic gene cluster, the environmental conditions eliciting its production/secretion and its mode of action. Recently, different genes (*fub1*, *fub2*, *fub3*, *fub4* and *fub5*) have been described as being responsible for FA biosynthesis in plant pathogenic *Fusarium* species, and their transcription has been found to be regulated by several environmental factors, including ambient pH, nitrogen source, nutrient availability and presence of a plant host [24]. Although the toxicity mechanism of FA is not fully understood, an increasing body of evidence suggests that metal chelation represents a major mechanism for the toxic effect of FA on plants, mammals and competing rhizobacteria [13,15]. In this work, we investigated the effect of environmental conditions on FA production by *Fol*. Previous work by Lopez-Berges and co-workers [25] showed that *Fol* is able to use a large variety of nitrogen sources. While a readily metabolized source ammonium inhibited virulence-related functions, a non-preferred source nitrate promoted such functions [25]. Moreover, it was shown that the ability of fungal pathogens to invade and kill plants depends upon cellular iron homeostasis, environmental iron availability and rhizosphere pH [22,26,27,28,29]. High iron availability and acidic pH inhibit virulence, while low iron availability and alkaline pH promote infection [29].

Here, we found that *Fol* cultures grown in the presence of ammonium show acidic pH values, while those supplemented with nitrate or urea show high pH values. It is important to note that iron availability in soils depends largely on redox potential and pH, where iron solubility decreases as soil pH increases [30]. To overcome reduced iron availability, soil-inhabiting microbes have evolved a battery of high-affinity, low-molecular-weight iron chelators known as siderophores, which are secreted into the environment for efficient acquisition of limited iron pools [31]. In line with this, *Fol* cultures grown on nitrogen sources leading to alkaline pH values showed higher chelating activity than those grown in acidic conditions. Interestingly, a similar pattern was observed for FA accumulation in culture supernatants, suggesting a possible role in FA production under iron-limiting conditions. In support of this idea, external addition of the chelating agent EDTA or the bacterial siderophore pyoverdine, as well as buffering the medium to pH 7.0, induced an increase in FA levels, while exogenous addition of iron or buffering to pH 4.5 resulted in reduced FA production. Thus, FA biosynthesis in *Fol* appears to be regulated by environmental pH and, consequently, iron availability. In line with this, *Fol* mutants in PacC, which are affected in external pH sensing, were altered in FA production and chelating activity. This is in agreement with previous reports indicating the requirement of PacC for efficient expression of *fub* genes in *F. fujikuroi* and siderophore production in *Aspergillus nidulans* at alkaline pH [14,32]. It is noteworthy that, similarly to siderophore biosynthesis, FA production also appears to depend on iron homeostasis in *Fol* because deletion of the iron homeostasis regulator HapX [22], a repressor of siderophore biosynthesis [33], led to an overall increase in FA levels and chelating activity. Thus, FA production in *Fol* might be regulated via two independent mechanisms based on pH and iron sensing. 

In the banana pathogen *F. oxysporum* f. sp. *cubense,* Bck1, Mkk2 and Mpk1, three conserved elements of the CWI MAPK signalling cascade, were shown to positively regulate the expression of FA biosynthetic genes and FA production [16]. Here, we found that *Fol* mutants lacking Mpk1, which is required for host sensing and virulence [18], produced less FA, suggesting that FA production in *F. oxysporum* is regulated by the CWI MAPK pathway. Surprisingly, the α-pheromone specific receptor Ste2, which was recently shown to signal via the CWI MAPK pathway to regulate chemotropism and conidial germination in *Fol* [18,34,35], also showed lower FA accumulation, suggesting a new role of pheromone autocrine signalling in FA production. Importantly, this effect was independent of extracellular pH and chelating activity, indicating that pheromone signalling-mediated FA production functions downstream of pH and Fe sensing in *Fol*. 

Collectively, our results show that virulence-promoting conditions such as alkaline pH, low iron availability, poor nitrogen sources and CWI MAPK signalling are associated with an increase in FA production, suggesting that *Fol* has evolved both independent and overlapping strategies to fine-tune the production of this important mycotoxin. 

## 4. Materials and Methods 

### 4.1. Fungal Isolates and Culture Condition

Fungal strains derived from *Fusarium oxysporum* f. sp. *lycopersici* (*Fol*) isolate 4287 (FGSC 9935) and used in this study are reported in Table 2. For microconidia production, fungal strains were grown in Potato Dextrose Broth (PDB; Difco; Fisher Scientific; Rodano, MI, Italy) at 28 °C with shaking at 180 rpm for 5 days. For in vitro fusaric acid (FA) detection, Puhalla minimal medium (MgSO_4_·7 H_2_O 2 mM, KH_2_PO_4_ 7 mM, KCl 6 mM, Sucrose 90 mM) adjusted to pH 5.0 and supplemented with 25 mM of different nitrogen sources (NaNO_3_; (NH_4_)_2_SO_4_; NH_4_NO_3_; CH_4_N_2_O) was used [36]. Where indicated, the medium was pH buffered to 7.0 or 5.0 with 100 mM 4-morpholineethanesulfonic acid monohydrate (MES) or supplemented with 100 μM FeCl_3_, 0.2 mM EDTA or 0.2 mM pyoverdine. Fungal strains were inoculated in the growth medium at a final concentration of 5 × 10^5^ conidia mL^−1^ and incubated for 7 days at 28 °C on a rotary shaker regulated at 180 rpm. The dry weight of the fungal biomass and pH of the culture broth were evaluated periodically. 

### 4.2. Chrome Azurol S Assay

Siderophore quantification in fungal supernatants was carried out by using the liquid chrome azurol S assay (CAS) (as previously described [37]. The percentage of chelating activity (CA%) was indirectly quantified by measuring the OD_655_ of the culture supernatant (ODs) and the uninoculated medium (ODc), used as a control, 60 min after the start of the reaction, with the following formula: CA% = [(ODc − ODs)/ODc] × 100.

### 4.3. Fusaric Acid Extraction and Quantification

For FA extraction from fungal cultures, supernatants were collected and filtered through Whatman no. 4 filter paper (Whatman Ltd., Maidstone, UK), adjusted to pH 2.0 with HCl (37% v/v) and extracted with ethyl acetate (1:1 v/v). Ethyl acetate phases were separately collected, dried by rotary evaporation under reduced pressure and resuspended in methanol. FA content was quantified through high-performance liquid chromatography (HPLC-UV; Varian Analytical Instruments; Model 9010; Palo Alto, CA, USA) by using previously reported methods and experimental conditions [38,39] and expressed in µg of FA per mg of dry fungal biomass. To obtain an FA calibration curve, methanol-dissolved FA standard solution (Merck Life Science; Milano, MI, Italy) was injected at concentrations ranging from 0.02 to 2.0 mg mL^−1^. A linear relationship between peak areas and the investigated FA concentrations was obtained (Y = 0.001x + 0.0105; R^2^ value of 0.997). Method validation was performed by spiking MM blanks with known concentrations of FA. The average recovery rate was 96% and always exceeded 95%.

### 4.4. Statistical Analyses

Data were submitted to variance analysis (ANOVA) using the SPSS software v. 16.0 (SPSS Inc., Chicago, IL, USA) and means compared with Tukey’s test. Before analyses, percentages of chelating activity were converted into Bliss angular values (arcsine square root of the percentage value). All experiments were repeated at least three times, with similar results. Homogeneity of variance for independent repetitions of each experiment was tested, and data from separate experiments having homogeneous variances were pooled.

## Figures and Tables

**Figure 1 toxins-15-00050-f001:**
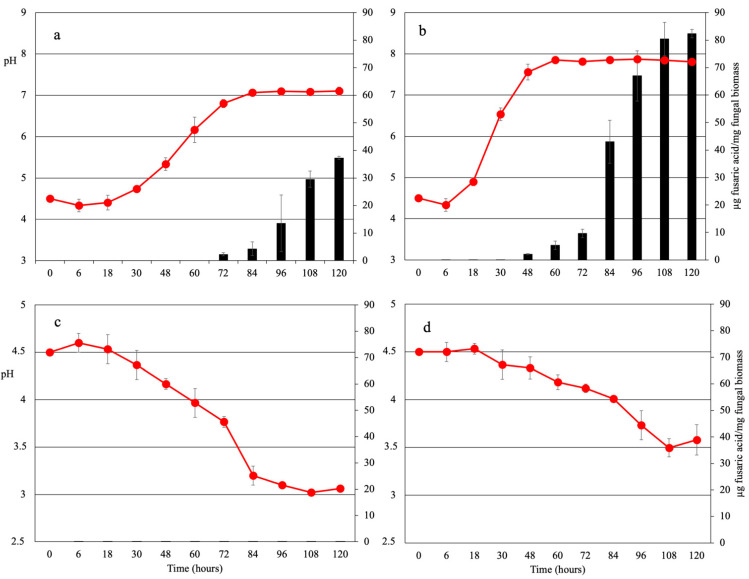
Trend of pH (red line) and fusaric acid concentration (black bars) in the culture supernatants of *Fusarium oxysporum* f. sp. *lycopersici* grown in minimal medium supplemented with urea (**a**), sodium nitrate (**b**), ammonium sulphate (**c**) or ammonium nitrate (**d**) as sole nitrogen sources. Bars represent standard deviations from three independent replicates. Experiments were performed three times, with similar results.

**Figure 2 toxins-15-00050-f002:**
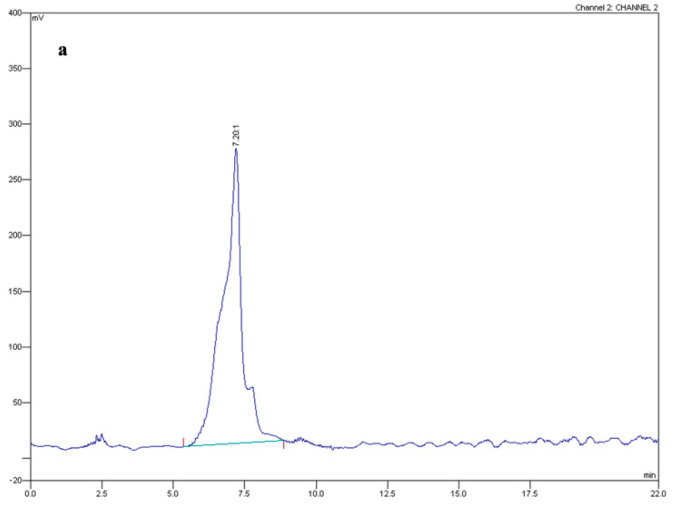
HPLC chromatograms obtained by injecting 20 µg mL^−1^ of a fusaric acid standard (**a**) or an ethyl acetate fraction obtained from *Fusarium oxysporum* f. sp. *lycopersici* cultures grown in the presence of urea (**b**), sodium nitrate (**c**), ammonium sulphate (**d**) or ammonium nitrate (**e**) as sole nitrogen sources. The red vertical lines in the graph mark the start and end of each peak.

**Figure 3 toxins-15-00050-f003:**
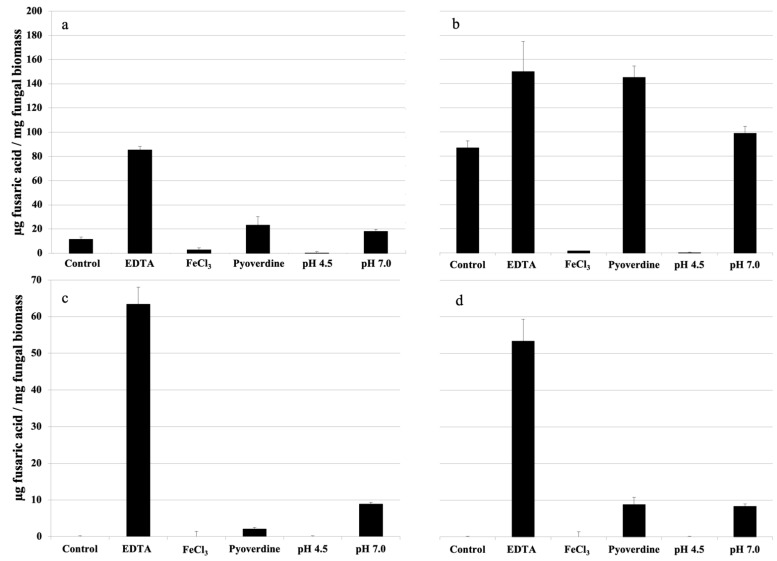
Fusaric acid content in the culture filtrates of *F. oxysporum* f. sp. *lycopersici* grown in minimal medium containing urea (**a**), sodium nitrate (**b**), ammonium sulphate (**c**) or ammonium nitrate (**d**) as sole nitrogen sources and supplemented with 0.2 mM EDTA, 100 µM FeCl_3_ or 0.2 mM pyoverdine, or pH buffered with 100 mM MES to pH 4.5 or 7. Bars represent standard deviations from three independent replicates. Experiments were performed three times, with similar results.

**Table 1 toxins-15-00050-t001:** Effect of different nitrogen sources on extracellular pH modification, chelating ability and fusaric acid content in the culture supernatants of the indicated *F. oxysporum* f. sp. *lycopersici* (*Fol*) strains. For each fungal strain, values marked by common letters are not different according to Tukey’s test (*p* ≤ 0.05). For each nitrogen source values marked with the * symbol are statistically different (*p* ≤ 0.05; according to Tukey’s test) from wild-type values. The values in the table are averages of three independent experiments with three replicates each.

*Fol* Strain	Nitrogen Source	pH			Chelating Activity			Fusaric Acid ^1^		
Wild type	Urea	7.02	a		78.00	a		37	a	
	Sodium nitrate	7.81	b		68.40	b		82	b	
	Ammonium sulphate	3.07	c		27.55	c		n.d.	c	
	Ammonium nitrate	3.64	d		25.55	c		n.d.	c	
*fmk1*Δ	Urea	6.94	a		75.78	a		41	a	
	Sodium nitrate	7.85	b		69.46	b		75	b	
	Ammonium sulphate	2.88	c		22.98	c		n.d.	c	
	Ammonium nitrate	4.16	d		23.90	c		n.d.	c	
*hog1*Δ	Urea	6.81	a		75.75	a		36	a	
	Sodium nitrate	7.12	b		65.37	b		85	b	
	Ammonium sulphate	2.93	c		25.55	c		n.d.	c	
	Ammonium nitrate	3.60	d		25.16	c		n.d.	c	
*mpk1*Δ	Urea	8.95	a	*	89.05	a	*	12	a	*
	Sodium nitrate	8.22	b	*	95.16	a	*	42	b	*
	Ammonium sulphate	3.01	c		21.09	b		n.d.	c	
	Ammonium nitrate	4.16	d		23.51	b		n.d.	c	
*ste2*Δ	Urea	8.40	a	*	92.05	a	*	9	a	*
	Sodium nitrate	8.33	a	*	91.16	a	*	39	b	*
	Ammonium sulphate	3.07	b		28.09	b		n.d.	c	
	Ammonium nitrate	3.92	c		27.51	b		n.d.	c	
*bck1*Δ	Urea	8.80	a	*	90.59	a	*	11	a	*
	Sodium nitrate	8.52	a	*	92.13	a	*	45	b	*
	Ammonium sulphate	2.94	b		21.09	b		n.d.	c	
	Ammonium nitrate	3.91	c		24.76	b		n.d.	c	
*mkk2*Δ	Urea	8.71	a	*	93.93	a	*	8	a	*
	Sodium nitrate	8.35	a	*	93.02	a	*	62	b	*
	Ammonium sulphate	3.12	b		28.43	b		n.d.	c	
	Ammonium nitrate	4.00	c		23.46	b		n.d.	c	
*hapX*Δ	Urea	7.05	a		84.80	a	*	63	a	*
	Sodium nitrate	7.89	b		86.41	a	*	70	a	
	Ammonium sulphate	3.02	c		60.56	b	*	27	b	*
	Ammonium nitrate	3.98	d		66.75	b	*	0.22	b	*
*pacC^c^*	Urea	7.18	a		77.89	a		37	a	
	Sodium nitrate	7.77	b		67.77	b		50	b	*
	Ammonium sulphate	3.12	c		58.28	c	*	22	c	*
	Ammonium nitrate	4.06	d		56.28	c	*	12	d	*
*pacC*Δ	Urea	7.08	a		35.55	a	*	n.d.	a	
	Sodium nitrate	7.58	b		32.25	a	*	n.d.	a	
	Ammonium sulphate	3.05	c		28.35	b		n.d.	a	*
	Ammonium nitrate	4.18	d		29.25	b		n.d.	a	*

^1^ Fusaric acid concentration is expressed in µg of the compound per mg of dry fungal biomass. n.d.= not detectable.

**Table 2 toxins-15-00050-t002:** *Fusarium oxysporum* f. sp. *lycopersici* (*Fol*) wild-type and mutant strains used in the experiments.

*Fol* Strain	Genotype	Gene Function	Reference
FGSC 4287	Wild type		[19]
*fmk1*Δ	*fmk1::PHLEO*	MAPK	[19]
*mpk1*Δ	*mpk1::HYG*	MAPK	[18]
*hog1*Δ	*hog1::HYG*	MAPK	[17]
*ste2*Δ	*ste2::HYG*	GPCR	[18]
*mkk2*Δ	*mkk2::HYG*	MAPKK	[18]
*bck1*Δ	*bck1::HYG*	MAPKKK	[18]
*hapX*Δ	*hapX::HYG*	Transcription factor	[22]
*pacC*Δ	*pacC::HYG*	Transcription factor	[21]
*pacC^C^*	*pacC^C^::HYG*	Transcription factor	[21]

## Data Availability

Not applicable.

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
