# Peer review of "Alkaline pH, Low Iron Availability, Poor Nitrogen Sources and CWI MAPK Signaling Are Associated with Increased Fusaric Acid Production in Fusarium oxysporum"

_toxins, 2023, doi:10.3390/toxins15010050_

Round 1
Author Response
1) The title of the paper is not comprehensive enough to extract the core content of the research and cannot accurately reflect the conclusion.
Reply: We thank the reviewer for the valuable suggestion. We have rephrased the title of the paper as follows: Alkaline pH, low iron availability, poor nitrogen sources and CWI MAPK signaling are associated with increased fusaric acid production in Fusarium oxysporum
2) In the abstract, the author mentions that "In plants, metal chelation is considered one of the possible mechanisms of action." This view needs to be supplemented with corresponding references.
Reply: Done. We have now added a reference to support this statement.
3) In the abstract, the author should be more concise and clearer about the research conclusions, and explain the necessity of the research.
Reply: We are grateful for the suggestion. We have now generated a more focused abstract with a clearer outline of research conclusions and importance of the research findings. The new abstract reads as follows:
Fusaric acid (FA) is one of the first secondary metabolites isolated from phytopathogenic fungi belonging to the genus Fusarium [1] This molecule exerts a toxic effect on plants, rhizobacteria, fungi and animals, and plays a crucial role in both plant and animal pathogenesis [2]. In plants, metal chelation by FA is considered one of the possible mechanisms of action [2]. Here we evaluated the effect of different nitrogen sources, iron content, extracellular pH and cellular signalling pathways on the production of FA siderophores by the pathogen Fusarium oxysporum (Fol). Our results show that nitrogen source affects iron chelating activity and FA production. Moreover, alkaline pH and iron limitation boost FA production while acidic pH and iron sufficiency repress it independent of nitrogen source. FA production is also positively regulated by the cell wall integrity (CWI) mitogen activated protein kinase (MAPK) pathway and inhibited by the iron homeostasis transcriptional regulator HapX. Collectively, this study demonstrates that factors promoting virulence (i.e. alkaline pH, low iron availability, poor nitrogen sources and CWI MAPK signalling) are also associated with increased FA production in Fol. The obtained new insights on FA biosynthesis regulation can be used to prevent both Fol infection potential and toxin contamination.
4) Other minor mistakes:
L219,there is no spaces between words;
Reply: Done. Now the title reads as follows: Chrome Azurol S assay
Fig 1, there is no solid line of coordinate axis;
Reply: Done. A thicker solid line for coordinate axis has been added to Fig.1 graphs.
Fig 3, the horizontal line in the coordinate axis should be deleted;
Reply: We disagree with the reviewer on this point. The horizontal line on the coordinate axis of the two upper panels of Fig.3 is required to indicate where graphs and bars in top panels begin.
Reviewer 2 Report
I found the manuscript very interesting, in general is well structured, the background of this study is sufficiently described and related to work done previously by other scientists. The results obtained as well as the conclusion are sound and concise and very interesting considering the importance of fusaric acid production by Fol.
I have done a minimal number of questions, that the authors need to addressed, throughout the whole paper in the attached pdf.
I will recommend the publication of the paper after minor revision.

Author Response
1) tracheomycosis (What?):
Reply: The term tracheomycosis refers to a fungal disease which causes xylem disfunction in plants as it is the case for fungi belonging to the Fusarium oxysporum species complex.
2) The x axys title is misssing. It is time?
Reply: We thank the reviewer for the helpful comment. We have now modified the graph by inserting the appropriate x axis title [i.e. Time (hours)] in the revised version of the manuscript.
Reviewer 3 Report
This manuscript is clear and well written and addresses a real problem.
However, I have one major criticism of this work. The main part of the study is based on the HPLC analysis of Fusaric acid. The validation of the method used is not given, the references on which this method is based are not present (ref 41) or are not complete enough in terms of validation (ref 40).
The levels of fusaric acid reported in this study vary according to the culture conditions. In order to prove that it is not the culture media that changes the extraction and assay results, it is necessary to present recovery data with spiked samples under these different study conditions.
What is the exact range of calibration (see below)? Is the method linear? What is the recovery?
Figure 1 : add legend for x-axis
Figure 2: it is said that 20µg/mL were injected but in the Material and methods L233 it is said that calibration range was 0.2-2mg/mL, please clarify
Spiked samples should be done to confirm the result
L231: there is no ref 41 and what was said in ref 40 is not enough to validate a method.
Reviewer 4 Report
My comments can be found in the attached MS.

Author Response
1) This sentence is incomplete-- possible meachnisms of action for the removal of FA or what?
Reply: Metal chelation is considered one of the mechanisms of action of FA in plants. To clarify this concept, we have now rephrased the indicated sentence as follows: In plants, metal chelation by FA is considered one of the possible mechanisms of action.
2) F. oxysproum infects animals and humans too?Did you mean to refer to FA?
Reply: F. oxysporum is considered an emerging animal and human pathogen. It is indeed recognized as a multihost model to genetically dissect fungal virulence mechanisms in both plants and mammals. Please see the following papers:
- a) Ortoneda M, Guarro J, Madrid MP, et al. Fusarium oxysporum as a multihost model for the genetic dissection of fungal virulence in plants and mammals. Infect Immun. 2004;72(3):1760-1766. doi:10.1128/IAI.72.3.1760-1766.2004
- b) Papri N, Sathi P, Surbhi S, Sampa D. Defence response in plants and animals against a common fungal pathogen, Fusarium oxysporum. Current Research in Microbial Sciences 2022; 3: 100135, ISSN 2666-5174, https://doi.org/10.1016/j.crmicr.2022.100135.
Round 2
Reviewer 1 Report
i think the ms could be accepted.
Reviewer 3 Report
Thank you for these methodological clarifications.
I have only one more very minor formal remark: It is not usual to see bibliographical references in an abstract, please check if this is allowed by the journal.